# COVID-19 vaccine hesitancy: Vaccination intention and attitudes of community health volunteers in Kenya

**Joachim Osur** [1,2]*, **Evelyne Muinga**[2], **Jane Carter** [2], **Shiphrah Kuria** [2], **Salim Hussein** [3], **Edward Mugambi Ireri** [1,4]

**1** Amref International University, Nairobi, Kenya, **2** Amref Health Africa Headquarters, Nairobi, Kenya, **3** Ministry of Health, Nairobi, Kenya, **4** Smart Health Consultants Limited Company, Nairobi, Kenya

* joachim.osur@Amref.org

**Data Availability Statement:** The data associated with the manuscript has been deposited on the Mendeley database: https://data.mendeley.com/datasets/j78w4s258j/1.

## Abstract

In Kenya, community health volunteers link the formal healthcare system to urban and rural communities and advocate for and deliver healthcare interventions to community members. Therefore, understanding their views towards COVID-19 vaccination is critical to the country's successful rollout of mass vaccination. The study aimed to determine vaccination intention and attitudes of community health volunteers and their potential effects on national COVID-19 vaccination rollout in Kenya. This cross-sectional study involved community health volunteers in four counties: Mombasa, Nairobi, Kajiado, and Trans-Nzoia, representing two urban and two rural counties, respectively. COVID-19 vaccination intention among community health volunteers was 81% (95% CI: 0.76–0.85). On individual binary logistic regression level, contextual influence: trust in vaccine manufacturers (adjOR = 2.25, 95% CI: 1.06–4.59; $p$ = 0.030); individual and group influences: trust in the MoH (adjOR = 2.12, 90% CI: 0.92–4.78; $p$ = 0.073); belief in COVID-19 vaccine safety (adjOR = 3.20, 99% CI: 1.56–6.49; $p$ = 0.002), and vaccine safety and issues: risk management by the government (adjOR = 2.46, 99% CI: 1.32–4.56; $p$ = 0.005) and vaccine concerns (adjOR = 0.81, 90% CI: 0.64–1.01; $p$ = 0.064), were significantly associated with vaccination intention. Overall, belief in COVID-19 vaccine safety (adjOR = 2.04, 90% CI: 0.92–4.47 $p$ = 0.076) and risk management by the government (adjOR = 1.86, 90% CI: 0.94–3.65; $p$ = 0.072) were significantly associated with vaccination intention. Overall vaccine hesitancy among community health volunteers in four counties in Kenya was 19% (95% CI: 0.15–0.24), ranging from 10.2−44.6% across the counties. These pockets of higher hesitancy are likely to negatively impact national vaccine rollout and future COVID-19 vaccination campaigns. The determinants of hesitancy arise from contextual, individual and group, and vaccine or vaccination specific concerns, and vary from county to county.

## Introduction

Coronavirus disease 2019 (COVID-19) caused by the novel coronavirus, severe acute respiratory syndrome coronavirus 2 (SARS-CoV-2) was first detected on 17 November, 2019, in

**Funding:** The funding source was American Tower Corporation. https://atckenya.ke/en/index.html. All authors were beneficiaries of the grant. The funders had no role in study design, data collection and analysis, decision to publish, or preparation of the manuscript. No authors received a salary from the funders. This grant does not have a funding number. It was a donation from a corporation, which was not won through a competitive process, so there's not even a call for proposals (or something like that) and thus no grant number.

**Competing interests:** The authors declare no conflict of interest.

Wuhan City, Hubei Province, China, and rapidly spread across the globe. On 30 January, 2020, the World Health Organization (WHO) declared the novel coronavirus outbreak a public health emergency of international concern (PHEIC), WHO's highest level of alarm. COVID-19 has resulted in numerous deaths worldwide, and one of the main strategies to control the pandemic is mass vaccination. Several SARS-CoV-2 vaccines have been developed including viral vector vaccines: Johnson & Johnson (Janssen COVID-19), AstraZeneca (Oxford) and Sputnik V (Gamaleya); protein-based vaccines (Novavax); mRNA vaccines (Pfizer-BioNTech and Moderna); and inactivated virus vaccines: Sinopharm, CoronaVac (Sinovac) and Covaxin (Bharat Biotech).

The WHO Strategic Advisory Group of Experts (SAGE) on Immunisation defines vaccine hesitancy as the delay in accepting or refusing vaccines despite the availability of vaccination services [1]. Several studies have been published on predictors of COVID-19 vaccination intention and vaccine hesitancy. In the United States of America (USA), Khubchandani *et al.* [2] reported gender, education, employment, income, having children at home, political affiliation and perceived COVID-19 threat as predictors of vaccine hesitancy. A further study in the USA by Fridman *et al.* [3] reported mistrust about the government withholding COVID-19 information and lack of honesty. In Canada, Dzieciolowska *et al.* [4] reported gender, age and occupational exposure as predictors of COVID-19 vaccination intention. Two studies in Brazil by Oliveira *et al.* [5] and Bivar *et al.* [6] reported females, older adults, evangelicals, lack of COVID-19 symptoms; and scepticism of the genuine interest of industry and politicians, lack of trust in research, and inaccurate COVID-19 information on social media as predictors of vaccine hesitancy, respectively. A multicountry study in South America by Argote *et al.* [7] reported side effects, fast development of vaccines, government mistrust and uncertainty of vaccine effectiveness as predictors of COVID-19 vaccine hesitancy. In China, Wang *et al.* [8] reported suspicion of the efficacy of COVID-19 vaccines, vaccine effectiveness and safety, beliefs that vaccines are unnecessary and lack of time to take vaccination as the main predictors of vaccine hesitancy. In Hong Kong, Kwok *et al.* [9] reported age, more confidence in COVID-19 vaccination, less complacency, and more collective responsibility as predictors of vaccination intention; and vaccine effectiveness, side effects and the duration of vaccine efficacy as predictors of COVID-19 vaccine hesitancy. In Israel, Dror *et al.* [10] reported self-perception of high risk for severe COVID-19 disease and gender as predictors of COVID-19 vaccination intention, while not being associated with or not caring for COVID-19 positive patients as predictors of vaccine hesitancy. In Saudi Arabia, Qattan *et al.* [11] reported gender, perceived high risk of COVID-19 infection, and perception of mandatory vaccination for all as predictors of COVID-19 vaccination intention. In a multicountry study in the Middle East, Sallam *et al.* [12] reported COVID-19 misinformation, conspiracy beliefs, reliance on social media for COVID-19 vaccine information, gender and education levels as predictors of vaccine hesitancy. In Australia, Edwards *et al.* [13] and Rhodes *et al.* [14] reported religiosity, populist views, gender, age and income; and vaccine efficacy, vaccine safety, and belief that vaccines are unnecessary as predictors of COVID-19 vaccination intention, respectively. A study by Dodd *et al.* [15] reported inadequate health literacy and lower education level as predictors of vaccine hesitancy. In Italy, Di Gennaro *et al.* [16] reported Facebook as the primary source of COVID-19 information, and being a non-physician, vaccine safety, and receiving little or conflicting information about COVID-19 vaccines as predictors of vaccine hesitancy. In the United Kingdom (UK), Razai *et al.* [17] reported ethnicity, education level; and Robertson *et al.* [18] reported ethnicity, education level, vaccine side effects, unknown future vaccine effects and lack of trust in vaccines as predictors of vaccine hesitancy. Paul *et al.* [19] reported mistrust of vaccine benefits and concerns about future unforeseen side effects; and Freeman *et al.* [20] reported age, gender, income level, ethnicity, lower adherence to social distancing

guidelines, vaccine efficacy, side effects, and speed of COVID-19 vaccine development as predictors of vaccine hesitancy. In a study in the UK and Ireland, Murphy *et al.* [21] reported gender, age and income level as predictors of vaccine hesitancy. In France, Gagneux *et al.* [22] reported that nurses and assistant nurses were more hesitant than physicians. On the African continent, in the Democratic Republic of the Congo (DRC), Nzaji *et al.* [23] reported a positive attitude as the predictor of COVID-19 vaccination intention; in Egypt, Fares *et al.* [24] reported the absence of adequate clinical trials, fear of side effects, vaccine safety, and healthcare cadres as predictors of vaccine hesitancy.

Community health workers are community members with basic training to promote health or carry out limited healthcare services but are generally not healthcare professionals [25]. In Kenya, community health workers are volunteers who do not officially draw a monthly income, although they may be paid a stipend as a motivating factor and compensation for their time. One of the main roles of community health volunteers (CHV) in Kenya is to support child health and immunisation, provide counselling on immunisation schedules, mobilise communities during immunisation days, identify and refer children for immunisation, trace and refer defaulters, and assist in immunisation campaigns [26]. Therefore, it is expected that CHVs will play a major role in convincing communities in Kenya to accept COVID-19 vaccination.

The objective of the current study was to determine COVID-19 vaccination intention and the possible determinants of vaccine hesitancy amongst CHVs in four counties in Kenya. Addressing the scope of COVID-19 vaccine hesitancy is an initial step towards building trust in COVID-19 vaccination efforts [27]. In the current study, vaccination intention referred to willingness to get vaccinated once COVID-19 vaccines are available; this definition differs from vaccine hesitancy which refers to delay in acceptance or refusal of vaccines despite availability of vaccine services. Based on systematic literature reviews, interviews with immunisation managers and WHO SAGE recommendations, attitudes as determinants of vaccine hesitancy were grouped into contextual influences, individual and group influences, and vaccine or vaccination specific issues. Contextual influences address historical, socio-cultural, environmental, health system and institutional, economic and political factors; individual and group influences address personal perceptions of vaccination and influences of the social and peer environment; and vaccine safety directly addresses issues related to specific vaccines or the vaccination process. This study addressed vaccine hesitancy by assessing these three groups of influences as potential determinants of vaccine hesitancy on the uptake of COVID-19 vaccination by CHVs in four counties in Kenya.

## Materials and methods

### Ethics statement

The Amref Ethics and Scientific Review Committee approved the research protocol and tools (ESRC P940/2021). The National Commission for Science, Technology and Innovation issued the research licence (NACOSTI-P-21-9428). Since the survey tools were administered virtually, verbal informed consent was sought from all study participants. The interviewers read out the informed consent form (S1 File) to the participants, ensured their understanding, and then signed the informed consent form (S1 File) on their behalf upon being given verbal consent to continue with the interviews.

### Study design

A cross-sectional study design was employed to determine contextual, individual and group influences, and vaccine-specific issues on vaccination intention. The study was conducted in

four counties in Kenya based on purposive sampling according to level of urbanisation and a range of low-income, middle-income, and high-income areas. Nairobi and Mombasa represented cosmopolitan populations mainly living in closely clustered urban settlements; Trans Nzoia is the agricultural basket of Kenya, with a widely dispersed population; and Kajiado represented the pastoralist Maasai, the primary livestock herders in the country, with a dispersed, nomadic population.

## Target population

Registered CHVs from the four counties were selected from lists of CHVs within each county's health departments. CHVs were included if they were attached to a community unit with a link health facility, and had practised for at least one year. Other cadres of community health workers were excluded from this study. Data were collected between 22–27 March, 2021.

## Sample size determination

At the time of conducting the study, there were 11,064 registered CHVs in the study sites: Nairobi 5,620, Mombasa 2,291, Kajiado 1,448 and Trans Nzoia 1,705. With an already known population, the sample size was determined using the Krejcie and Morgan (1970) formula:

$$s = \chi^2 NP(1 - P) \div d^2(N - 1) + \chi^2 P(1 - P)$$

Where
s = required sample size
$X^2$ = the table value of chi-square for 1 degree of freedom at the desired confidence level $p$ = 0.05 (3.841)
$N$ = the population size (11,064)
$P$ = the population proportion (assumed to be 0.50 since this would provide the maximum sample
$d$ = with the degree of accuracy expressed as a proportion (0.05)
The formula is based on $p$ = 0.05, assuming the probability of committing a type 1 error was less than $p < 0.05$. The total sample size was 371 plus an additional 10% adjustment for non-response.

A proportional stratified sampling method was then used to determine the sample of CHVs within the four counties. To ensure adequate representation of both urban and rural counties and recognising that urban counties have more sub-counties, the number of sub-counties studied was standardised to four per county, which was the lowest number of sub-counties in the four sampled counties. R script programming was then used to randomly choose the CHVs proportionately to the population size (number of CHVs) of each county: Trans Nzoia 63 (15%), Kajiado 53 (13%), Nairobi 207 (51%) and Mombasa 85 (21%) which gave a minimum sample size of 408 CHVs.

## Data collection tool

The survey questionnaire was derived, adapted and conceptualised from the WHO SAGE vaccine hesitancy matrix [28]. The questions focused on investigating the attitude of CHVs on COVID-19 vaccines and were set in English, one of Kenya's official languages. Quantitative data were collected through telephone interviews, while qualitative data were collected through key informant interviews which were conducted virtually to align with COVID-19 prevention measures. The S1 Questionnaire was divided into two sections. Section A focused on demographic variables: gender, age, religion, level of education, years of services as a CHV, number

of households under the supervision of the CHV, attending a Ministry of Health (MOH) approved COVID-19 training course, income, and involvement in educating the community on COVID-19. Section B focused on questions addressing vaccination intention and attitude. Attitude was investigated on three levels: contextual influences, individual and group influences, and vaccine safety and vaccine-specific issues.

The survey questions on contextual influences tested communication and the media environment; influential leaders, gatekeepers and anti- or pro-vaccination lobbies; historical influences; religion/culture/gender/socio-economic parameters; politics/policies; geographical barriers and the pharmaceutical industry. Individual and group influences tested experience with past vaccination; beliefs, attitudes about health and prevention; knowledge and awareness; health system and provider trust and personal experience; risks and benefits; and immunisation as a social norm. Vaccination specific issues were tested using scientific evidence, the introduction of new vaccines or new formulations; mode of administration; design of the vaccination programme and mode of delivery, and reliability or source of vaccine supply.

## Data collection process

The following sub-counties were sampled: in Nairobi, Westlands, Lang'ata, Kasarani and Embakasi West; in Mombasa, Changamwe, Kisauni, and Nyali; in Trans Nzoia Kwanza and Sabaoti; and in Kajiado, Kajiado Central and Kajiado East. MOH officials from the national government and county governments approved the study and supported the data collection process. In each sub-county, a public health official was recruited as a research assistant and trained virtually for one day; a total of 11 public health officials were trained and assisted in the data collection process. In line with protocols for combating the spread of COVID-19 disease, telephone interviews were adopted to collect the survey data.

## Statistical analysis

**Training and test data sets.** Data were collected from the sample of 413 CHVs. A vector of indices having an 80% random sample was subset using the Maecheler *et al's* [29] Caret package in R where the dataset was split into a train ($n = 330$) and a test ($n = 83$) dataset based on a ratio of 80:20, respectively. Data associated with this study is found at https://data.mendeley.com/datasets/j78w4s258j/1.

**Fitting logistic regression.** COVID-19 vaccination intention was the dependent variable while attitude, vaccine safety and vaccine-specific issues, contextual influence, individual and group influences were independent variables. A hierarchical binary logistic regression model was created using only the significant independent variables. Four logistic regression models were used: model1 investigated trust in vaccine manufacturers; model2 investigated trust in the MOH and belief in vaccine safety; model3 investigated risk management of COVID-19 vaccine side effects, trust in health systems, and concerns on safety of COVID-19 vaccines; and Model4 investigated a combination of the above three models under one model. The odds ratios and *p* values were calculated using the [30] Visreg package in R.

## Model fit statistics

**Test of model adequacy using the linktest.** The final model4 adequacy was estimated by generating linktest *hat* (the predicted value) and *hatsq* (the predicted value squared) to determine if the model was properly specified. The *hat* should be highly significant, and evidence of a good fit is reflected in a non-significant *hatsq*. The values for model1 were hat ($p = 0.029$) and hatsq ($p = NA$); model2 hat ($p = 0.601$) and hatsq ($p = 0.497$); model3 hat ($p = 0.083$) and

**Table 1. McFadden pseudo R-squared.**

|  | Model1 | Model2 | Model3 | Model4 |
|---|---|---|---|---|
| McFadden | 0.01 | 0.08 | 0.08 | 0.10 |
| Cox and Snell | 0.01 | 0.07 | 0.07 | 0.09 |
| Nagelkerke | 0.02 | 0.11 | 0.12 | 0.15 |
| Likelihood.ratio.test | 0.036 | 0.0000042 | 0.000013 | 0.000015 |

hatsq ($p = 0.669$); and model4hat ($p = 0.016$) and hatsq ($p = 0.571$) values which suggested model adequacy.

**Hosmer and Lemeshow goodness of fit test.** The evidence of overall goodness of fit is reflected in a non-significant $p$-value. The Hosmer and Lemeshow goodness of fit was estimated using the [31] ResourceSelection package. The non-significant Hosmer and Lemeshow goodness of fit test were model1 $\chi^2$ (7) = 5.5e-26, $p = 1$; model2 $\chi^2$ (7) = 1.1e-26, $p = 1$; model3 $\chi^2$ (7) = 1.21, $p = 0.99$, and model4 $\chi^2$ (7) = 1.92, $p = 0.96$ all of which were evidence of overall goodness of fit.

**Nagelkerke, McFadden pseudo R-squared, Cox and Snell, nagelkerke, and likelihood ratio test.** Pseudo-r-squared statistics were used to explain the variation in the four models. The r-squared value was not used because there are situations where better models have lower r-squared statistics. The nature of the dataset and the number of predictor variables affect the value of the R squared. Table 1 shows the statistics used to test the fit of the model. Four models were tested and thus the McFadden pseudo-R-squared was used instead of the r-squared statistics used in linear regression because its value increases as the model's fit improves; equally, it is used to compare the fit of multiple models to the same data set. The findings in Table 1 show that the McFadden pseudo R-squared improves with the addition of predictor variables. According to McFadden [32], for maximum likelihood estimation, values of 0.2 to 0.4 represent an excellent fit. Thus, the models used in this study were promising.

**Average marginal effects computation.** The impact of the predictors on the outcome variables was described by estimating the marginal effect, which is the change in outcome as a function of the change in the independent variable holding all other variables in the model constant. The average marginal effects for the three models were computed using the [33] margins package. The average marginal effects of the independent variables were contextual influences; trust in vaccine manufacturers (0.128; CI:95% 0.014–0.240; $p = 0.027$); individual and group influences; vaccine safety (0.169; CI:95%, 0.071–0.269; $p = 0.0008$), and vaccine-specific issues; risk management (0.131; CI:95%, 0.044–0.291; $p = 0.0033$). This indicated that for a 1-unit increase in trust in vaccine manufacturers, trust in vaccine safety, and trust in risk management, the probability of CHVs' vaccination intention will significantly increase by 12.8%, 16.9% and 13.1%, respectively.

## Model evaluation on test data

After fitting the binary logistic model, the next step was to check how well the fitted model performs on the unseen 20% of the test data.

**Predict the test dataset.** A binary classification with a cut-off value of 0.5 was set. Any value below 0.5 on vaccination intention was considered negative (0), while 0.5 and above was considered positive (1). A confusion matrix between the actual (neg:0, pos:1) and the predicted (neg:0, pos:1) values was created, and classification of accuracy was determined. The confusion matrix showed that the test dataset on model1 had 12 sample cases of negative (0) and 71 cases of positive (1), model2 had 12 sample cases of negative (0) and 69 cases of positive (1), model3 had 11 sample cases of negative (0) and 71 cases of positive (1), and model4 had 11 sample cases of negative (0) and 70 cases of positive (1).

**Prediction of the classification accuracy using the unseen test data.** The classification accuracy of the final model was estimated using the [34] yardstick package. The accuracy of 0.855 shows that the classifier was about 85.5% accurate in classifying the unseen 20% test data.

The preliminary analysis focused on Chi-square statistics while hierarchical binary logistic regression was conducted. The crude odds and adjusted odds ratios were interpreted. The tables were plotted using the Stargazer package in R [35], the statistical analyses were performed using R script programming [36], and the graphs were plotted using the ggplot2 package [37].

# Results

## Sociodemographic

The total number of CHVs interviewed was 413, representing Nairobi 209 (50.6%), Mombasa 84 (20.3%); Kajiado 54 (13.1%); and Trans Nzoia 66 (16%). Table 2 shows the segregated demographics, and Fig 1 shows the significant independent variables.

## Inferential statistics

Overall, 267 (81%) CHVs intended to receive COVID-19 vaccination once made available in the country (95% CI: 0.76–0.85), and 63 (19%) were hesitant (95% CI: 0.15–0.24). There was a wide variation in COVID-19 vaccination intention between the counties. The distribution of those who intended to receive vaccination was: Nairobi 184 (44.6%), Mombasa 49 (11.9%), Kajiado 42 (10.2%) and Trans Nzoia 60 (14.5%); the distribution of those who did not intend to receive vaccination was Nairobi 25 (6.1%), Mombasa 35 (8.5%), Kajiado 12 (2.9%) and Trans Nzoia 6 (1.5%), $\chi^2$ (3) = 39.52, $p < 0.001$. Vaccination intention showed significant differences based on level of education and county. Nairobi County showed a significant association between level of education and vaccine acceptance, $\chi^2$ (2) = 6.70, $p = 0.035$, with the acceptors tending to be among the better educated. This association in Kajiado County was not conclusive because the Chi statistics result failed to meet the minimum threshold of interpreting the significant value. There was no significant association between level of education and vaccine intention in Mombasa and Trans Nzoia Counties. There was significantly higher vaccination intention among CHVs exposed to MoH approved training on COVID-19 (62.5%) even if not trained specifically on vaccination (18.6%), $\chi^2$ (1) = 5.56, $p = 0.018$. Thus, the factors significantly associated with vaccination intention among CHVs were based on county of origin (region), level of education, and previous exposure to MoH approved training on COVID-19.

**Attitude: Contextual influences.** There was a significant association between intention to accept vaccines and cultural opposition to COVID-19 vaccination, $\chi^2$ (1) = 4.68, $p = 0.030$; trust in MoH decisions on COVID-19 vaccination, $\chi^2$ (1) = 13.36, $p < 0.001$; and trust in good intentions of vaccine manufacturers, $\chi^2$ (1) = 9.82, $p = 0.002$. Based on regions, Nairobi stood out as significantly different from other counties on trust in good intentions of vaccine manufacturers, $\chi^2$ (1) = 8.47, $p = 0.004$; and trust in MoH decisions on COVID-19 vaccination, $\chi^2$ (1) = 7.01, $p = 0.008$. In Kajiado County, trust in MoH decisions was significantly associated with acceptance of COVID-19 vaccination; $\chi^2$ (1) = 4.19, $p = 0.041$. Binary logistic regression on contextual influences indicated that trust in good intentions of vaccine manufacturers was the only independent variable that had a significant effect on vaccination intention (adjOR = 2.25, 95% CI: 1.06–4.59; $p = 0.030$) as shown in Table 3.

The alluvial diagram in Fig 2 shows that most CHVs believed that COVID-19 vaccine manufacturers had good intentions. However, almost an equal percentage of CHVs with positive

**Table 2. Demographic characteristics.**

| Demographic | Nairobi | Mombasa | Kajiado | Trans Nzoia |
|---|---|---|---|---|
| Sex | Male; 43 (20.6%) | Male; 20 (23.8%) | Male; 20 (37%) | Male; 25 (37.9%) |
| | Female; 164 (78.5%) | Female; 63 (75%) | Female; 33(61.1%) | Female; 41 (62.1%) |
| | Other; 2 (1%) | Other; 1 (1.2%) | Other; 1 (1.9%) | |
| Age (Years) | 18–24; 15 (7.2%) | 18–24; 8 (9.5%) | 18–24; 5 (9.3%) | 18–24; 3 (4.5%) |
| | 25–35; 56 (26.8%) | 25–35; 28 (33.3%) | 25–35; 26 (48.1%) | 25–35; 6 (9.1%) |
| | Above 35; 138 (66%) | Above 35; 48 (57.1%) | Above 35; 23 (42.6%) | Above 35; 57 (86.4%) |
| Religion | Catholic; 65 (31.1%) | Catholic; 18 (21.4%) | Catholic; 7 (13%) | Catholic; 8 (12.1%) |
| | Protestant; 103 (49.3%) | Protestant; 28 (33.3%) | Protestant; 45 (83.3%) | Protestant; 57 (86.4%) |
| | Islam; 7 (3.3%) | Islam; 28 (33.3%) | Islam; 2 (3.7%) | Islam; 1 (1.5%) |
| | Others; 34 (16.3%) | Others; 10 (11.9%) | | |
| Education | Primary; 58 (27.8%) | None; 2 (2.4%) | None; 1 (1.9%) | Primary; 13 (19.7%) |
| | Secondary; 108 (51.7%) | Primary; 25 (29.8%) | Primary; 20 (37%) | Secondary; 47 (71.2%) |
| | Others; 43 (20.6%) | Secondary; 29 (34.5%) | Secondary; 21 (38.9%) | Others; 6 (9.1%) |
| | | Others; 28 (33.3%) | Others; 12 (22.2%) | |
| Years of service | less than 3; 29 (13.9%) | less than 3; 22 (26.2%) | less than 3; 15 (27.8%) | less than 3; 5 (7.6%) |
| | 3–5; 49 (23.4%) | 3–5; 16 (19%) | 3–5; 16 (29.6%) | 3–5; 4 (6.1%) |
| | Above 5; 131 (62.7%) | Above 5; 46 (54.8%) | Above 5; 23 (42.6%) | Above 5; 57 (86.4%) |
| Households | less than 20; 2 (1%) | less than 20; 12 (14.3%) | less than 20; 14 (25.9%) | less than 20; 3 (4.5%) |
| | 21–50; 14 (6.7%) | 21–50; 25 (29.8%) | 21–50; 22 (40.7%) | 21–50; 12 (18.2%) |
| | Above 50; 193 (92.3%) | Above 50; 47 (56%) | Above 50; 18 (33.3%) | Above 50; 51 (77.3%) |
| COVID-19 MOH Training | | Not sure; 3 (1.4%) | Not sure; 5 (6%) | No; 16 (24.2%) |
| | No; 20 (9.6%) | No; 32 (38.1%) | No; 13 (24.1%) | Yes; 34 (51.5%) |
| | Yes; 186 (89%) | Yes; 47 (56%) | Yes; 41 (75.9%) | Not sure; 16 (24.2%) |
| Educating | No; 9 (4.3%) | No; 19 (22.6%) | No; 4 (7.4%) | No; 25 (37.9%) |
| | Yes; 200 (95.7%) | Yes; 65 (77.4%) | Yes; 50 (92.6%) | Yes; 41 (62.1%) |
| Income | CHV; 79 (37.8%) | CHV; 51 (60.7%) | CHV; 15 (27.8%) | CHV; 33 (50%) |
| | OFE; 9 (4.3%) | OFE; 10 (11.9%) | OFE; 5 (9.3%) | |
| | ONFE; 121 (57.9%) | ONFE; 23 (27.4%) | ONFE; 34 (63%) | ONFE; 33 (50%) |

Notes: OFE = Other Formal Employment, ONFE = Other Non-Formal Employment, CHV Years = Years of Service as a Community Health Volunteer, Income = Your main source of Income, educating = Have you been involved in educating the community on COVID-19? Training = Have you had MoH approved training on COVID-19? Households = Number of households you are responsible for.

responses had no intention of getting vaccinated. Equally, many CHVs who did not trust COVID-19 manufacturers had the intention of receiving vaccination once vaccines were made available.

**Attitude: Individual and group influences.** There was a significant association on intention to accept vaccination and the feeling that information on COVID-19 vaccines is being openly shared; $\chi^2 (1) = 5.13$, $p = 0.023$; trust in what the MoH says about COVID-19 vaccination, $\chi^2 (1) = 18.96$, $p < 0.001$; belief in COVID-19 vaccine safety, $\chi^2 (1) = 20.51$, $p < 0.001$; and support for mass COVID-19 vaccination, $\chi^2 (1) = 16.20$, $p < 0.001$. Mombasa County was significantly different from the other counties in trusting the MOH on COVID-19 vaccination; $\chi^2 (1) = 4.70$, $p = 0.030$. Binary logistic regression indicated only two independent variables had positive significant effects on vaccination intention: trust in what the MoH says about COVID-19 vaccination (adjOR = 2.12, 90% CI: 0.92–4.78; $p = 0.073$), and belief in COVID-19 vaccine safety (adjOR = 3.20, 99% CI: 1.56–6.49; $p = 0.002$), as shown in Table 3, and the alluvial diagrams in Figs 3 and 4.

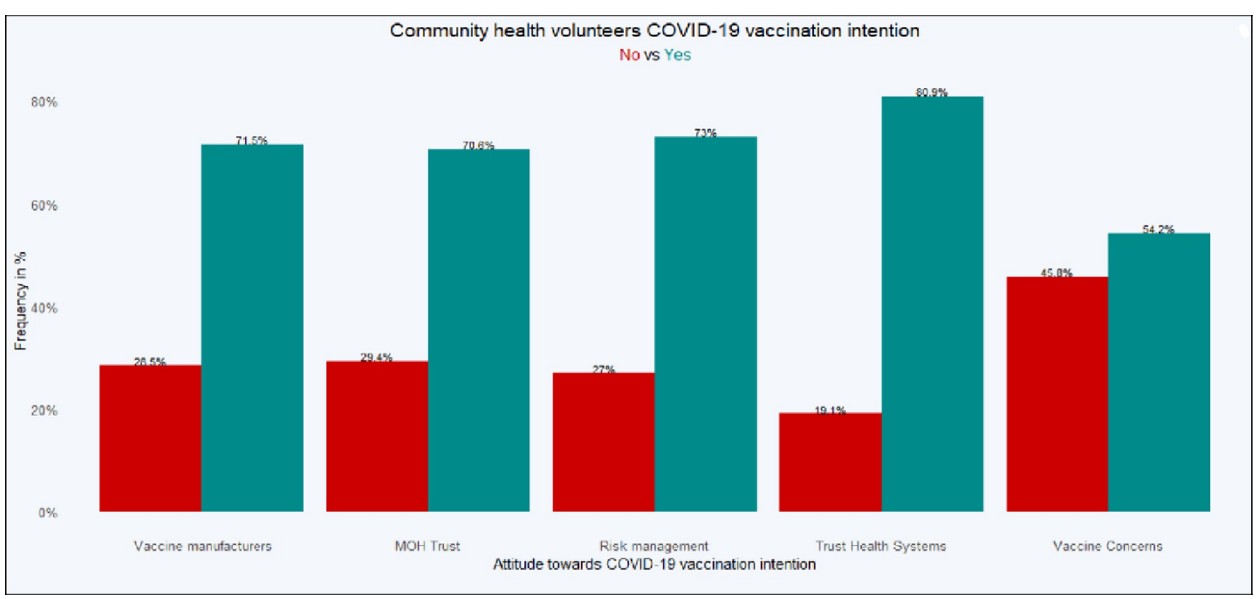

**Fig 1. COVID-19 vaccination intention among community health volunteers.**

**Table 3. Attitude on COVID-19 vaccination intention (hierarchical binary logistic regression).**

|  | Dependent variable: COVID-19 vaccination intention | | | |
|---|---|---|---|---|
|  | Beta estimates and $p$ values | | | |
| **Independent variables** | **logit model** | **logit model** | **logit model** | **logit model** |
|  | **(1)** | **(2)** | **(3)** | **(4)** |
| (1). Trust vaccine manufacturers | 0.809 |  |  | 0.376 |
|  | $p = 0.030^{**}$ |  |  | $p = 0.378$ |
| (2). Trust Ministry of Health |  | 0.752 |  | 0.384 |
|  |  | $p = 0.073^{*}$ |  | $p = 0.415$ |
| (3). Belief in vaccine safety |  | 1.163 |  | 0.714 |
|  |  | $p = 0.002^{***}$ |  | $p = 0.076^{*}$ |
| (4). Risk management of COVID-19 vaccine side effects |  |  | 0.899 | 0.623 |
|  |  |  | $p = 0.005^{***}$ | $p = 0.072^{*}$ |
| (5). Trust in health systems |  |  | 0.618 | 0.147 |
|  |  |  | $p = 0.130$ | $p = 0.754$ |
| (6). Concerns on the safety of COVID-19 |  |  | -0.215 | -0.170 |
|  |  |  | $p = 0.064^{*}$ | $p = 0.152$ |
| Constant | 0.693 | -0.124 | 0.937 | 0.19 |
|  | $p = 0.042^{**}$ | $p = 0.717$ | $p = 0.089^{*}$ | $p = 0.768$ |
| Observations | 330 | 330 | 330 | 330 |
| Log Likelihood | -162.935 | -152.749 | -152.457 | -149.067 |
| Akaike Inf. Crit. | 329.87 | 311.498 | 312.914 | 312.134 |

Note

$^{*}p < 0.1$

$^{**}p < 0.05$

$^{***}p < 0.01$.

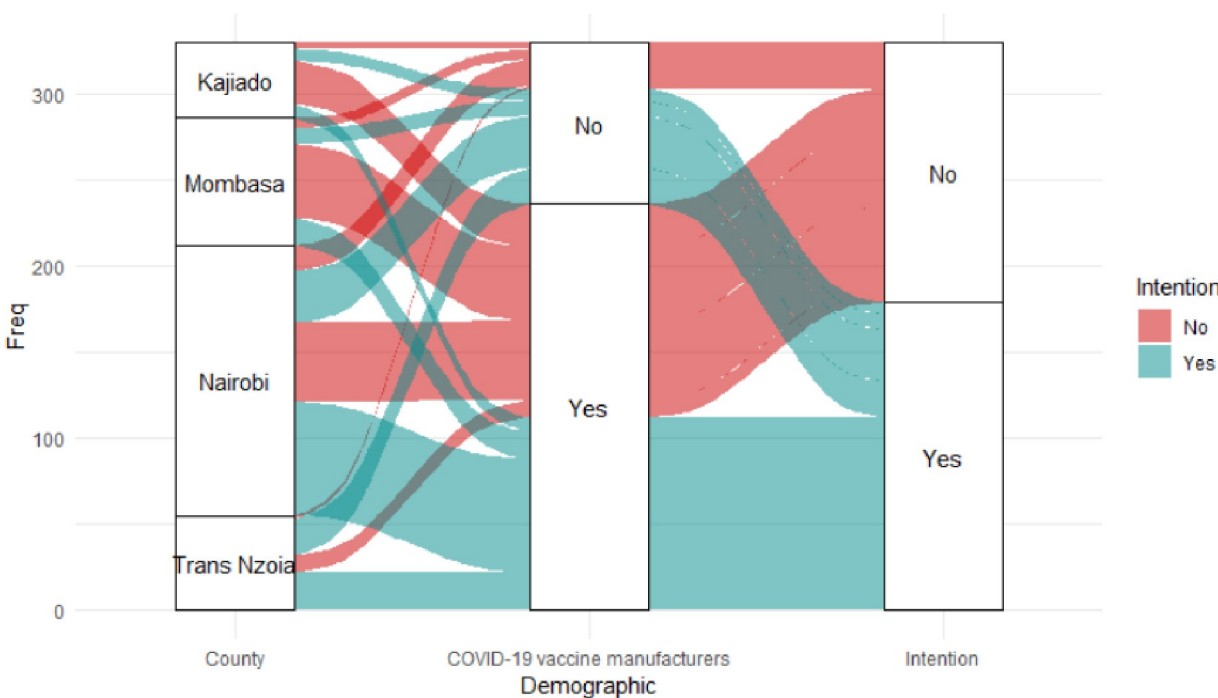

Do you think vaccine manufacturers have good intentions for you and people in your community?

**Fig 2. Trust in good intention of vaccine manufacturers.**

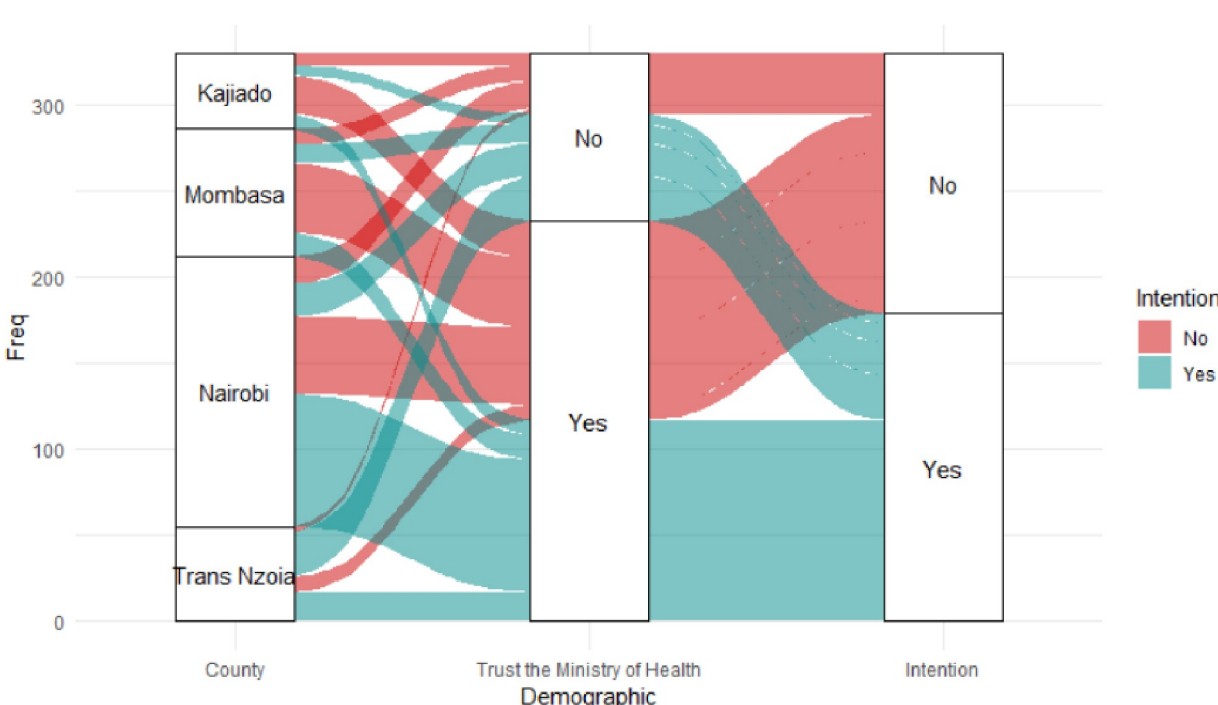

Do you trust what the MoH says about COVID-19 vaccination?

**Fig 3. Trust the MOH on COVID-19 vaccination.**

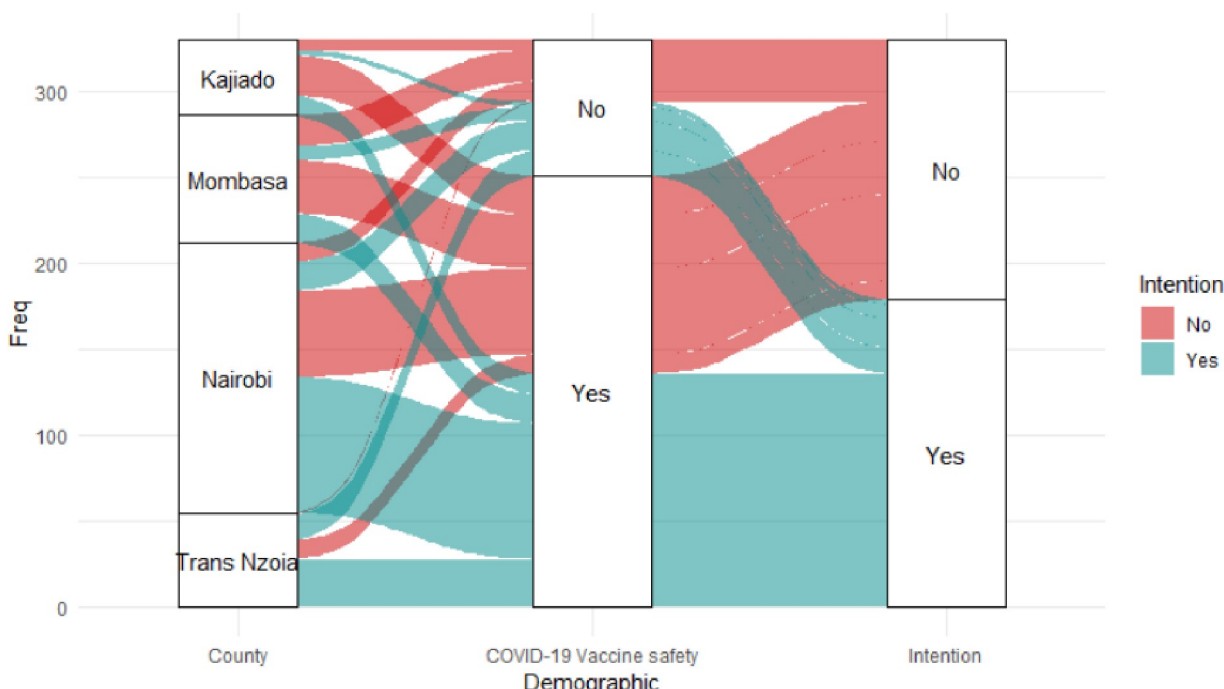

**Fig 4. Belief in vaccine safety.**

Fig 3 shows that most CHVs in Nairobi had the intention of receiving vaccination although a large proportion of them had no intention of getting vaccinated. Mombasa County had a large proportion of CHVs unwilling to get vaccinated although they had trust in the MoH, and belief in the safety of COVID-19 vaccines (Fig 4).

**Attitude: Vaccine safety and vaccination specific issues.** There was a significant association between intention to accept vaccination and the ability of the government to manage risks associated with COVID-19 vaccine side effects being openly shared, $\chi^2 (1) = 24.60, p < 0.001$; trust in the health system to deliver COVID-19 vaccination to communities, $\chi^2 (1) = 20.06$, $p < 0.001$; confidence in the safety of COVID-19 vaccines, $\chi^2 (1) = 23.43, p < 0.001$; concerns that COVID-19 vaccines might not be safe for the public, $\chi^2 (1) = 19.55, p = 0.001$; and the overall feeling about the safety of COVID-19 vaccines for the general population $\chi^2 (1) =$ 13.76, $p = 0.008$. Nairobi stood out as significantly different from the other counties on the belief that the government can manage risks associated with vaccination, $\chi^2 (1) = 9.54$, $p = 0.002$. A significant majority of CHVs in Nairobi trusted that the health system could deliver vaccines to the community, $\chi^2 (1) = 10.68, p = 0.001$. However, the significance level was not reliable due to the violation of Chi statistics interpretation measures. Binary logistic regression showed the following factors had significant positive effects on vaccination intention: the ability of the government to manage risks associated with COVID-19 vaccine side effects being openly shared (adjOR = 2.46, 99% CI: 1.32–4.56; $p = 0.005$) as shown in Table 3, and the alluvial diagrams in Figs 5 and 6.

Fig 5 shows that a large proportion of CHVs in Nairobi County believed in the government's COVID-19 risk management and intended to receive vaccination compared with those who did not believe in the government's COVID-19 management and had no intentions of getting vaccinated.

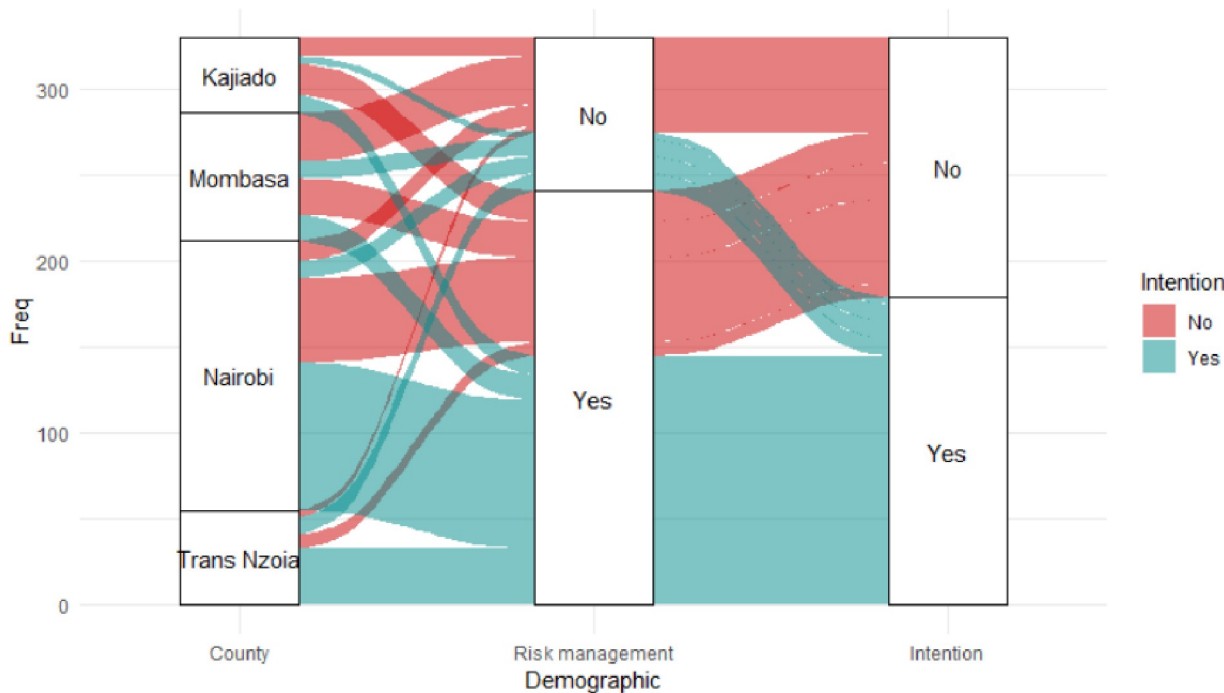

**Fig 5. Government COVID-19 risk management.**

Fig 6 shows that a large proportion of CHVs who had trust in the health systems had no intentions of getting vaccinated. The majority who believed in the health systems came from Nairobi county, followed by Mombasa county, and then Kajiado and Trans Nzoia respectively.

Negative significant effects were recorded: concerns that COVID-19 vaccines might not be safe for the public (adjOR = 0.81, 90% CI: 0.64–1.01; *p* = 0.064), as shown in Table 3 and the alluvial diagram in Fig 7.

A large proportion of CHVs from Trans Nzoia had no concerns at all about COVID-19 vaccine safety and contributed the second largest proportion after Nairobi for CHVs intending to get vaccinated. Although they intended to get vaccinated, the CHVs from Mombasa had the largest proportion of respondents who had no concerns about vaccine safety. Nairobi County led among the CHVs who were not concerned about vaccine safety and were intending to get vaccinated. The diagram (Fig 7) also shows that many CHVs were greatly concerned about COVID-19 vaccine safety and were not intending to get vaccinated.

The hierarchical binary logistics regression (model4) showed that contextual influences, and individual and group Influences had no significant effect on vaccination intention at the multivariate level. Nevertheless, COVID-19 risk management and vaccine safety significantly affected vaccination intention at the multivariate level: 'Do you feel our country can manage risks associated with COVID-19 vaccine side effects?' (adjOR = 1.86, 90% CI: 0.94–3.65; *p* = 0.072); 'In your view is the COVID-19 vaccine safe enough for people to be injected?' (adjOR = 2.04, 90% CI: 0.92–4.47 *p* = 0.076.

## Discussion

Most published studies on the prevalence of COVID-19 vaccination hesitancy have focused on healthcare workers with formal employment or the general population. This is the first study

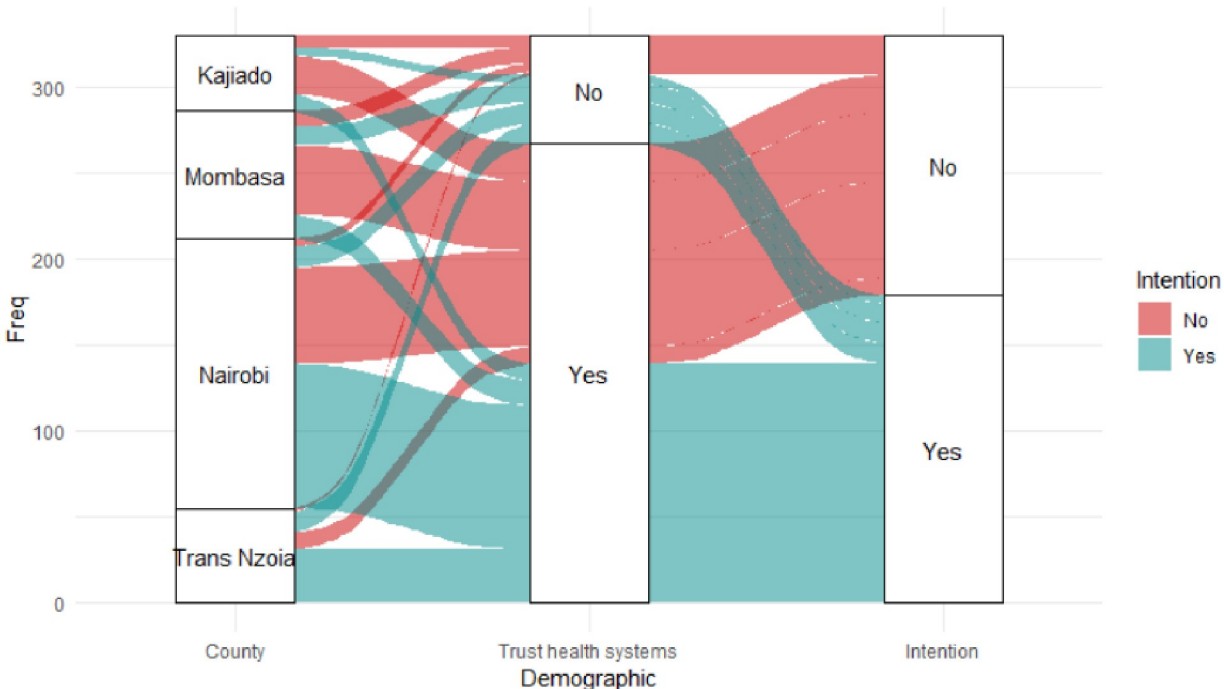

Can the health system be trusted to deliver the COVID-19 vaccine to your communities?

**Fig 6. Trust in health systems.**

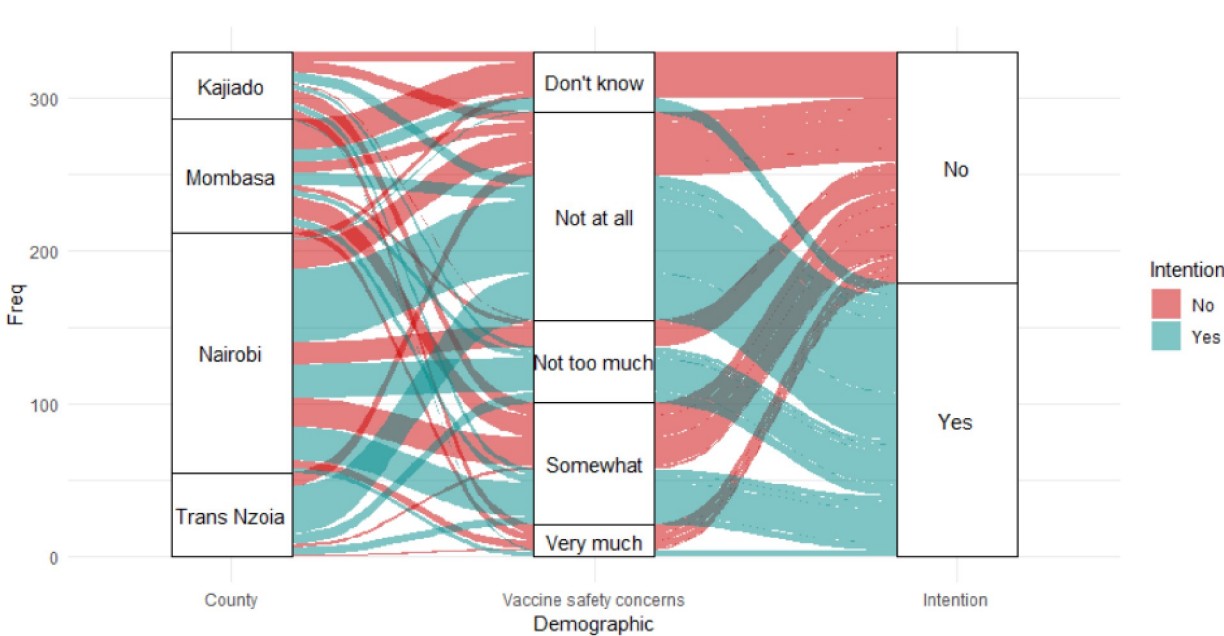

How concerned are you that the COVID-19 vaccine might not be safe for the public?

**Fig 7. Concerns about COVID-19 safety.**

to focus on COVID-19 vaccine hesitancy among community health workers in the African context. Community health workers link the form al healthcare system to communities in many low and middle-income countries (LMICs), including in Africa.

Insignificant socio-demographic factors in vaccination intention were gender, contrary to studies in USA [2], Canada [4], Saudi Arabia [11, 12], Australia [13] and UK [21]; age, contrary to studies in Australia [13], and UK [21]; religion, contrary to a study in Australia [13]; years of service as a CHV, the number of households each CHV was attached to, and source of income, contrary to studies in USA [2], Australia [13] and UK [21].

Overall COVID-19 vaccination intention among CHVs in the four counties studied in Kenya was 81%, with hesitancy or rejection in 19%, which would be explained by the government's ability to manage risks, trust in the health system, no concerns for vaccine safety, and being well informed on the vaccine. This figure is not far from the 18% hesitancy reported in the UK [18], 22% in the USA [2], and 17.5% in Brazil [5]. The low vaccine hesitancy in Kenya is associated with increased level of education, with vaccine acceptors tending to be among the more educated members of the communities, especially in Nairobi. This finding is in line with a study in the USA [2]; studies in Jordan, Kuwait and Saudi Arabia [12]; a study in Australia [15]; and one in the UK [18], all of which showed education was a significant predictor of vaccine acceptance. However, there was wide variation in vaccination intention between the study counties based on level of education: education had a statistically significant effect on CHVs in Nairobi who intended to get vaccinated $\chi^2$ (2) = 6.7, $p$ = 0.035 ($n$ = 209; 88%). Although this effect was not significant in Mombasa (58%), Kajiado (78%) and Trans-Nzoia (91%) Counties, there were large variations in terms of those who intended to get vaccinated. Equally, vaccination intention was higher among CHVs exposed to MOH training on COVID-19 ($n$ = 258; 62.5%) even if not specifically trained on vaccination ($n$ = 77;18.6%). Studies in the USA [38]; Italy [16]; Jordan, Kuwait and Saudi Arabia [12]; Australia [15]; Brazil [6]; and Bangladesh [39] reported that COVID-19 information delivered via social media and Facebook in Italy [16] were predictors of vaccine hesitancy. Overall, in our study, television and radio remained the most important sources of information among CHVs: those intending to get vaccinated had a higher tendency of getting information from radio and television. In comparison, those not intending to get vaccinated were significantly ($p$ = 0.002) associated with obtaining information from social media and community meetings such as *barazas* (meetings with chiefs) which present the best way of communicating vaccine safety to the community. No significant differences were reported between counties on these issues.

Health workers have a huge role to play in ensuring a high uptake of COVID-19 vaccination in Africa. Community health workers are the gatekeepers to society and their hesitancy to vaccination is likely to negatively impact the roll-out of vaccine programmes. Frontline health workers, including community health workers, are the major conduit of correct health information [40] and thus any misinformation among them is likely to result in COVID-19 vaccine hesitancy in their communities.

## Attitude (contextual influence)

The perception of CHVs on manufacturers of COVID-19 vaccines having good intentions to its users was reported to significantly increase vaccine intention and lower hesitancy. Any form of mistrust in manufacturers of COVID-19 vaccines will lead to hesitancy, which is supported by several studies. A study on COVID-19 vaccine hesitancy in Brazil [6] reported scepticism about the true interest of the industry; respondents in studies from Argentina, Brazil, Chile, Colombia, Mexico and Chile [7] expressed their concerns about vaccines being developed too fast; a study in the UK [20] reported concerns on the speed of development of the

vaccines, and a study in Egypt [24] mentioned the absence of adequate clinical trials. This study in Kenya showed that if CHVs believe the manufacturers have good intentions for their community members, then vaccination intention would increase by 25% (adjOR = 2.25).

## Attitude (individual & group influences)

The study indicated that trust in what the MoH says about COVID-19 vaccines would increase vaccination intention by 28% (adjOR = 2.28) among CHVs in Kenya. A study in Latin America [11] reported government mistrust as a determinant of COVID-19 vaccine hesitancy. Equally, a study among black Americans [38] reported mistrust about the government withholding COVID-19 vaccine information as a promotor of vaccine hesitancy. Belief in COVID-19 vaccine safety would increase vaccination intention by 20% (adjOR = 3.20) among CHVs. These findings are supported by studies in Italy [16]; China [9] Hong Kong [9]; Australia [14]; the UK [18–20]; and Egypt [24], all which reported that issues with vaccine safety would lead to COVID-19 vaccine hesitancy.

## Attitude: Vaccine safety and vaccine-specific issues

Our study showed that if CHVs felt the government can manage risks associated with COVID-19 vaccine side effects, the probability of vaccination intention would increase by 46% (adjOR = 2.46). Studies in Latin America [7]; Hong Kong [9]; the UK [18–20]; and Egypt [24] all reported COVID-19 vaccine side-effects as a predictor of vaccine hesitancy. Therefore, both governments and vaccine manufacturers need to openly place all required information about vaccines in the public domain.

The health system plays a crucial role in delivering routine vaccinations and will also play a significant role in delivering COVID-19 vaccination. Our study showed ($n$ = 311;75.3%) that CHVs with trust in the healthcare system in their county had a significantly stronger intention to deliver COVID-19 vaccine in their communities, in contrast to a few (n = 24;5.8%) who did not trust the health system in their county but still intended to get vaccinated. In Kenya, nurses are the primary providers of health services in rural health facilities and in some health facilities in urban settings. Thus, trust in nurses as part of the health system is a good predictor of vaccination intention among CHVs. However, studies in France [22]; Canada [4]; China [8]; and Israel [10] have shown nurses were the most COVID-19 vaccine-hesitant among healthcare cadres. In Africa, in the DRC, in a study that excluded doctors [23], other healthcare workers were hesitant to receive COVID-19 vaccine. A study by Ditekemana *et al* in DRC [41] reported that being a healthcare worker was associated with a decreased willingness to get vaccinated against COVID-19. Thus, to lower vaccine hesitancy in Kenya below 19%, targeted COVID-19 information needs to be directed towards all healthcare workers, and future studies should also focus on COVID-19 vaccine hesitancy among healthcare workers in Kenya.

Concerns have emerged regarding the safety of COVID-19 vaccines among the public. The findings from this Kenya sample show that concerns about COVID-19 vaccine safety would decrease the probability of vaccination intention among the CHVs by 81% (adjOR = 0.81). It is worth noting that 91 (22%) of CHVs across the four counties who were not concerned about COVID-19 vaccines being safe for the public intended to get vaccinated. This figure was lower than the number of CHVs ($n$ = 114;27.6%) who were ambivalent about whether COVID-19 vaccines were safe or not for the public. Overall, 78 (18.9%) CHVs were significantly concerned and did not intend to get vaccinated compared to 335 (81.1%) who intended to get vaccinated. Thus, the proportion of those not intending to get vaccinated was found to rise significantly with the increase in level of concern about COVID-19 disease ($\chi^2$ (4) = 19.55, $p$ = 0.001).

## Strengths and limitations

COVID-19 vaccination intention and hesitancy are highly variable from one community to another. Complete generalization of the findings may be challenging, and thus, data analysis was split by counties. The study was the first one of its kind in the eastern and southern African region and thus there is need to replicate the study in these regions. The study was conducted during the COVID-19 lockdown and restriction of movement, and thus, telephone surveys were considered the most appropriate. Most CHVs in Kenya have mobile telephones increasing the likelihood of gathering a representative sample to complete the survey; besides, the entire data collection process was cost-effective. Short and precise questions were used for easier comprehension of the survey. However, Amref Health Africa is well-known throughout Kenya, and thus telephone surveys might have been mistaken for telemarketing, thus influencing the responses from the CHVs.

## Recommendations and programmatic implications

The current study had several programmatic implications. Trust in COVID-19 vaccine manufacturers and trust in the MOH on COVID-19 vaccination is very important. Studies have already shown that trust is an important predictor for both vaccination intention and vaccine hesitancy. The study recommends supporting the work of CHVs on COVID-19 to enhance vaccine roll-out and future COVID-19 vaccination campaigns through training, provision of resources, deployment, and performance monitoring.

This study recommends training CHVs on COVID-19 and its vaccines to reduce hesitancy, and funds for training must accompany the vaccines. Building government trust will be essential even in future mass vaccination programmes. There should be enhanced provision of information and engagement with CHVs to increase trust in MOH messaging around the vaccines. In Kenya, the health system response should be strengthened in counties outside Nairobi to earn the trust of CHVs that the system is capable of handling the national COVID-19 vaccination programme. This is especially important if the government can show that it can manage risks associated with COVID-19 vaccine side effects. Concerns on COVID-19 vaccine safety should be addressed through increased training of CHVs and provision of COVID-19 information through popular media and vernacular outlets.

## Conclusions

There is a strong significant positive effect between CHVs' vaccination intention and their trust in vaccine manufacturers, trust in the health ministry, belief in vaccine safety, and trust in the government on risk management of COVID-19 vaccine side effects. Equally, there is a robust negative effect of COVID-19 vaccine concerns on vaccination intention. The vaccination intention evidence was most substantial in Nairobi, making it different from the other counties. Therefore, it is expected that in counties with high hesitancy among CHVs, it will be more difficult to mobilise communities to accept COVID-19 vaccination.

## Supporting information

**S1 Questionnaire. Cross-sectional survey questions for community health volunteers.** (DOCX)

**S1 File. Informed consent form.** Informed consent for cross-sectional survey. (DOCX)

## Author Contributions

**Conceptualization:** Joachim Osur, Evelyne Muinga, Jane Carter, Shiphrah Kuria, Salim Hussein.

**Data curation:** Edward Mugambi Ireri.

**Formal analysis:** Edward Mugambi Ireri.

**Funding acquisition:** Joachim Osur.

**Investigation:** Joachim Osur, Edward Mugambi Ireri.

**Methodology:** Joachim Osur, Evelyne Muinga, Edward Mugambi Ireri.

**Project administration:** Evelyne Muinga.

**Resources:** Joachim Osur, Salim Hussein.

**Software:** Edward Mugambi Ireri.

**Supervision:** Evelyne Muinga.

**Validation:** Joachim Osur, Edward Mugambi Ireri.

**Visualization:** Edward Mugambi Ireri.

**Writing – original draft:** Edward Mugambi Ireri.

**Writing – review & editing:** Jane Carter, Shiphrah Kuria, Edward Mugambi Ireri.

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
