## [Decision Letter · Decision Letter 0]

16 Aug 2021

 PGPH-D-21-00389 COVID-19 vaccine hesitancy: vaccination intentions and attitudes of community health volunteers in Kenya PLOS Global Public Health

Dear Dr. Joachim Osur,

Thank you for submitting your manuscript to PLOS Global Public Health. After careful consideration, we feel that it has merit but does not fully meet PLOS Global Public Health’s publication criteria as it currently stands. Therefore, we invite you to submit a revised version of the manuscript that addresses the points raised during the review process.

The reviewers have raised the serious issues about the description of methodology and presentation of the results. Plos Global Public Health aims to publish the methodologically sound scientific research. Therefore, carefully address each of the reviewers comments.  

We look forward to receiving your revised manuscript.

Kind regards,

Roopa Shivashankar, MD, MSc

Academic Editor

Reviewers' comments:

Reviewer's Responses to Questions

**Comments to the Author**

1. Does this manuscript meet PLOS Global Public Health’s publication criteria? Is the manuscript technically sound, and do the data support the conclusions? The manuscript must describe methodologically and ethically rigorous research with conclusions that are appropriately drawn based on the data presented.

Reviewer #1: Partly

Reviewer #2: No

Reviewer #3: Partly

2. Has the statistical analysis been performed appropriately and rigorously?

Reviewer #1: Yes

Reviewer #2: No

Reviewer #3: Yes

3. Have the authors made all data underlying the findings in their manuscript fully available (please refer to the Data Availability Statement at the start of the manuscript PDF file)?

Reviewer #1: Yes

Reviewer #2: No

Reviewer #3: Yes

4. Is the manuscript presented in an intelligible fashion and written in standard English?

Reviewer #1: Yes

Reviewer #2: Yes

Reviewer #3: No

5. Review Comments to the Author

Reviewer #1: Appreciate the authors for addressing an important research question relevant to the current pandemic

Introduction

• Efficacy and technologies of all vaccines are not required to be discussed in detail.

• Instead of listing different names used for community health volunteer, their roles and responsibilities can be mentioned and include why their hesitancy needs to be studied?

• Kindly delete the table with literature review; rather discuss the key findings from previous studies in introduction.

Materials and methods

• There are 47 counties in Kenya. How these four were selected? Sampling method?

• Please mention the parameters considered for sample size calculation

• Data collection details are not provided. Who? Where? and How?

• How was it possible to complete data collection within few days?

• Kindly describe about the data collection tools.

• First bar chart needs to be improved. It should be numbered. Axis lables to be added.

Inferential statistics

• Distribution can be written after mentioning the prevalence with numbers.

• Table 2 can be restructured to make it less crowded. Demographic variables as rows and counties as columns.

• What is R2 value for the goodness of fit?

• What factors were considered for four models?

• Independent variable being a categorical variable, is it required to present beta-coefficients?

Discussion

• “Insignificant socio-demographic factors in vaccination intention were gender (contrary to [15] study in the USA; [19] study in Canada; [28] study in Saudi Arabia; [29] study in Saudi Arabia; [32]study in Australia; [39] and study in the UK); age (contrary to [31] in Bangladesh; [32] study in Australia; [39] study in the UK), religion (contrary to [32] study in Australia), years of service as a CHV, the number of households each CHV was attached to, and source of income (contrary to [15] study in the USA; [22] study in Colombia; [31] study in Bangladesh; [32] study in Australia;and [39] study in the UK)”. – need to be summarized and rewritten

• Strengths and limitations of the study are not mentioned. Kindly mention it.

Recommendations

• “This study recommends training CHVs on COVID-19 and its vaccines to reduce hesitancy, and funds for training must accompany the vaccines” the prevalence of vaccine acceptance is 81% which is very high. Therefore is the above recommendation appropriate?

Conclusion

• “A strong association exists between CHVs’ vaccination intention and their readiness to engage with communities.” I couldn’t see any data suggestive of your conclusion?! Please make conclusions based on your findings.

Reviewer #2: 1. The objectives of the study done were to find out vaccination intention, attitude of the Community Health Volunteers (CHVs) and their potential effect on vaccine roll out. The authors reported vaccination intention and factors associated with. From results section it appears that attitude was measured by factors associated with vaccination intention. Materials and methods section does not mention how attitude or potential effect on vaccine roll out was measured.

2. Nothing is written about data collection tool which was used to gather the requisite information. It appears some interview was done with study participants, but details of the data collection tool need to be mentioned including the mode of collection.

3. The authors have used Krejcie Morgan Formula for sample size calculation. This formulae, is used for sample size calculation for a finite population. Some assumptions are needed for which are, population size, confidence level, degrees of freedom, margin of errors. None of these were mentioned in the article.

4. There are seven figures in the article. The figures are not explanatory and may be presented in table form.

Reviewer #3: Dear Authors,

The COVID vaccination and its uptake is dependent on multiple factors. I am glad the authors have explored this important subject which is of great importance. However, there are lot many issues in the manuscript that must be addressed before publication.

Please find my comments on the manuscript:

1. Introduction:

a. The authors have talked about the spread of COVID in the first paragraph which sets platform for the study. The second paragraph and third paragraph which talks about the type of vaccine and their final details is unwarranted to this context.

b. What is not clear to me is the prevailing issues which led the authors to undertake this study-needs a strong justification backed with facts and figures

c. What was the magnitude of COVID morbidity and mortality in Kenya? Other African countries?

d. Was there equitable distribution of vaccines all over the continent? Countries?

e. “Based on systematic literature reviews and interviews with immunization managers, attitudes as determinants of vaccine hesitancy were grouped into contextual influences, individual and group influences, and vaccine or vaccination specific issues.” – Needs further explanation. Who and why did they adapt this approach?

f. There is lot of description on community health volunteers but lacks strong justification on why they were chosen.

g. Summary of the published studies in table? What was the purpose of getting it in introduction? The same could have been explained in sentences. Probably, the focus should have been African countries.

h. There is no research question. There are no objectives.

2. Methods

a. Study setting: Need to describe about the country Kenya and its counties. Literacy rate? Population? Access to health care facilities? Availability of vaccines?

b. Hypothesis?

c. What was the sampling frame? How was sampling done? Describe sample size calculation? Target population?

d. What were the tools for data collection? Who, when and how were they collected? How was data validation done? Pilot study? Training? Issues?

e. What data variables are collected? Operational definition for each of these variables needs to be specified.

f. Data analysis: Needs to be specific and assumptions made needs justification?

g. Why are results discussed in data analysis??

3. Results

a. The results are complex and poorly described. Though, alluvial diagrams are attractive they fail to convey the messages. The country should be described in totality rather than individual counties.

b. Table 3 and 4 are not appropriately structured. It seems to be a output window of analysis.

c. The result section needs to be re-written

4. Discussion

a. The discussion lacks focus. Though, the authors have compared the findings with other studies; they have not appropriately reasoned out the current findings

b. Programmatic implications on the study findings

c. Strengths and limitations of the study

d. It is advisable to restructure and rewrite the discussion section.

6. PLOS authors have the option to publish the peer review history of their article (what does this mean?). If published, this will include your full peer review and any attached files.

**Do you want your identity to be public for this peer review?** For information about this choice, including consent withdrawal, please see our Privacy Policy.

Reviewer #1: No

Reviewer #2: No

Reviewer #3: **Yes: **Sharath Burugina Nagaraja

---

## [Decision Letter · Decision Letter 1]

9 Dec 2021

PGPH-D-21-00389R1

COVID-19 vaccine hesitancy: vaccination intentions and attitudes of community health volunteers in Kenya

Dear Dr. OSur,

Thank you for submitting your manuscript to PLOS Global Public Health. After careful consideration, we feel that it has merit but does not fully meet PLOS Global Public Health’s publication criteria as it currently stands. Therefore, we invite you to submit a revised version of the manuscript that addresses the points raised during the review process.

Suggest to shorten the introduction, specifically the literature on vaccine hesitancy starting at page 4 last paragraph.Please check for language, grammer and punctuations through out. 

We look forward to receiving your revised manuscript.

Kind regards,

Roopa Shivashankar, MD, MSc

Academic Editor

Journal Requirements:

Additional Editor Comments (if provided):

Reviewers' comments:

Reviewer's Responses to Questions

**Comments to the Author**

1. If the authors have adequately addressed your comments raised in a previous round of review and you feel that this manuscript is now acceptable for publication, you may indicate that here to bypass the “Comments to the Author” section, enter your conflict of interest statement in the “Confidential to Editor” section, and submit your "Accept" recommendation.

Reviewer #1: All comments have been addressed

Reviewer #3: All comments have been addressed

2. Does this manuscript meet PLOS Global Public Health’s publication criteria? Is the manuscript technically sound, and do the data support the conclusions? The manuscript must describe methodologically and ethically rigorous research with conclusions that are appropriately drawn based on the data presented.

Reviewer #1: Yes

Reviewer #3: Yes

3. Has the statistical analysis been performed appropriately and rigorously?

Reviewer #1: Yes

Reviewer #3: Yes

4. Have the authors made all data underlying the findings in their manuscript fully available (please refer to the Data Availability Statement at the start of the manuscript PDF file)?

Reviewer #1: Yes

Reviewer #3: Yes

5. Is the manuscript presented in an intelligible fashion and written in standard English?

Reviewer #1: Yes

Reviewer #3: Yes

6. Review Comments to the Author

Reviewer #1: Thanks for addressing all the comments.

Reviewer #3: (No Response)

7. PLOS authors have the option to publish the peer review history of their article (what does this mean?). If published, this will include your full peer review and any attached files.

**Do you want your identity to be public for this peer review?** For information about this choice, including consent withdrawal, please see our Privacy Policy.

Reviewer #1: No

Reviewer #3: **Yes: **Sharath Burugina Nagaraja

---

## [Editor Report · Decision Letter 2]

1 Feb 2022

COVID-19 vaccine hesitancy: vaccination intentions and attitudes of community health volunteers in Kenya

PGPH-D-21-00389R2

Dear Dr Osur,

We are pleased to inform you that your manuscript 'COVID-19 vaccine hesitancy: vaccination intentions and attitudes of community health volunteers in Kenya' has been provisionally accepted for publication in PLOS Global Public Health.

Best regards,

Roopa Shivashankar, MD, MSc

Academic Editor